# Assessment of protocols for characterization of the human skin microbiome using shotgun metagenomics and comparative analysis with 16S metabarcoding

Florian Plaza Oñate,[1] Benoît Quinquis,[1] Florence Thirion,[1] Marine Gilles,[1] Christian Morabito,[1] Karine Valeille,[1] Richard Martin,[2] Bérengère Guidet,[3] Catherine Kern,[4] Sophie Pécastaings[4]

**ABSTRACT** The skin microbiome includes bacteria, fungi, and viruses, with composition varying significantly across body sites. Although 16S rRNA gene sequencing is common, it excludes non-prokaryotic taxa and offers limited functional data. Shotgun metagenomics provides broader taxonomic and functional insights but is challenging for low-biomass skin samples due to limited microbial DNA and high host contamination. In this study, we characterized the microbiome of the forehead and armpits in healthy individuals using shotgun metagenomics and assessed the strategies to improve sequencing success. We compared collection kits, DNA extraction protocols, and tested multiple displacement amplification (MDA). We found that sampling with D-Squame discs followed by an in-house DNA extraction protocol was the most effective combination to maximize DNA yields. MDA introduced significant compositional biases and is not recommended. Shotgun sequencing, without MDA, produced microbial compositions and diversity indices broadly consistent with 16S rRNA metabarcoding, although it showed discrepancies in the relative abundance of some genera. Consistent with prior studies, the armpit microbiome was dominated by *Staphylococcus spp.*, whereas the forehead microbiome was dominated by *Cutibacterium spp*. Critically, shotgun sequencing provided additional insights into viral and eukaryotic microorganisms and revealed the functional potential of microbial communities, demonstrating its clear advantages over 16S rRNA metabarcoding for comprehensive skin microbiome research.

**IMPORTANCE** With growing evidence of the role of microorganisms in maintaining healthy skin, accurately characterizing the skin microbiome remains a significant challenge. In this study, we demonstrate that shotgun sequencing, carried out with adapted wet lab protocols, provides deep insights into the microbiome composition of specific areas, such as the forehead or the armpits. Notably, it enables the characterization of fungi and viruses while offering direct functional insights into microbial communities, providing a clear advantage over 16S ribosomal RNA gene sequencing. Our findings highlight the potential of shotgun metagenomics as a powerful tool for comprehensive skin microbiome analysis. They emphasize the importance of tailored protocols for low-biomass samples, improving the reliability of shotgun sequencing and paving the way for more robust clinical studies focused on the skin microbiome.

**KEYWORDS** skin microbiome, metagenomics, 16S RNA sequencing

Colonization of the human skin by microorganisms has been recognized and studied since the beginning of microbiological research. Longtime considered a source of pathogens, it redeemed interest in the last decades with the development of new

**Peer Reviewers** Xinzhao Tong, Xi'an Jiaotong-Liverpool University, Suzhou, Jiangsu Province, China; Bärbel Ulrike Foesel, Helmholtz Zentrum München, Neuherberg, Germany

Address correspondence to Florian Plaza Oñate, florian.plaza-onate@inrae.fr.

The authors declare no conflict of interest.

See the funding table on p. 17.

techniques of study, along with the discovery of beneficial effects of some major skin commensals (1, 2).

Being a complex and heterogeneous surface, skin harbors a combination of bacteria, yeast, and viruses whose proportions and species vary across the different skin areas (3). Micro-environments created by pilosebaceous units, sweat glands, or the microtopography of the skin also increase the diversity of microorganisms fit to these specific conditions. Indeed, sebaceous skin areas (back, forehead, and scalp) are essentially colonized by microorganisms able to degrade lipids, such as *Propionibacteria* or *Malassezia spp*. In contrast, moist skin areas (armpits and antecubital creases) are characterized by a diverse microbiome, where *Staphylococci* and *Corynebacteria* are more prevalent (4, 5).

Considering the biases associated with traditional culture methodologies, which only account for viable and cultivable bacteria and struggle to detect those in very low proportions, next-generation sequencing methods have significantly improved microbiome analysis. More specifically, 16S rRNA gene sequencing (16S sequencing) has brought new light to the skin microbiome and helped better define specific dysbiosis involved in skin diseases like acne, rosacea, or psoriasis (6, 7). This widely used technique consists of the amplification and sequencing of the 16S ribosomal RNA encoding gene. Currently, partial sequencing that targets only a small part of the 16S rRNA gene (e.g., variable regions V1–V3) is the most commonly available and cheapest technique. However, sequenced amplicons are often too small to be accurately assigned to a given species, and consequently, the taxonomic annotation usually stops at the family or genus level. In contrast, full-length 16S sequencing allows high-resolution taxonomic annotation at the species level, and sometimes up to the strain level (8). However, 16S sequencing, independently of the amplicon length, is restricted to prokaryotes and precludes the analysis of the fungal and viral communities. Furthermore, it has been reported that the copy number of the 16S rRNA gene varies not only between species but also within strains of the same species (9). This variability can alter the estimated abundances of certain taxa, and the statistical methods proposed to mitigate this bias provide unsatisfactory results (10). Although universal, the PCR primers used to amplify the 16S rRNA gene can also have varying affinity from one species to another. Thus, DNA from certain bacteria may be amplified more efficiently than others, leading to an inaccurate representation of some species (11). Finally, as 16S sequencing focuses on a single gene family, direct characterization of microbial metabolic potential is impossible.

In contrast, shotgun metagenomics involves sequencing millions of short DNA fragments randomly drawn from a microbial community. Because it does not target a specific gene family or region, this approach enables the detection and quantification of bacteria and archaea, as well as non-prokaryotic microorganisms such as fungi and viruses. Since metagenomic sequencing theoretically provides access to all genes present in a sample, it not only enables precise taxonomic profiling down to the species level but also offers insights into the metabolic potential of the microbial community under study. Furthermore, because it does not require targeted amplification of microbial DNA, metagenomic sequencing is less likely to introduce taxonomic biases than amplicon-based methods.

However, using metagenomic sequencing to characterize low-biomass microbiomes presents technical challenges. First, the absence of an amplification step limits its applicability to low-biomass samples such as those from the skin. The bacterial density on the skin generally ranges from $10^4$ to $10^6$ CFU/cm² (12), whereas human fecal samples contain between $10^9$ and $10^{12}$ CFU/mL (13). Consequently, DNA yields after extraction are often well below the levels recommended by manufacturers for sequencing library preparation, and whole genome amplification (WGA) is frequently required (14). During sample collection, microbial cells are detached from the skin surface, but decaying human corneocytes can significantly contaminate samples. As a result, the proportion of non-microbial DNA originating from the host genome can be high. In the worst-

case scenario, such as dry or damaged skin, host contamination exceeds 90% of the sequencing data generated (15).

Given the clear advantages of shotgun sequencing, the objective of this study was to optimize sample preparation and library construction to identify the best methodology for shotgun analysis of skin samples. Two different skin areas were analyzed: a sebaceous area (forehead) and a moist area (armpit). In parallel, samples were analyzed by 16S sequencing (partial V1–V3 and full length) to compare the outputs of both approaches on biologically relevant samples characterized by low microbial biomass and high levels of host DNA. To our knowledge, this is the first comparison of metabarcoding and shotgun metagenomics of human skin samples.

## RESULTS

### Sampling and extraction protocols impact DNA yields

Forty healthy volunteers were enrolled in the study. Four samples were collected from each participant: two from the forehead and two from the armpit. Samples were initially intended either for shotgun metagenomic sequencing (120 samples from 40 individuals) or for 16S rRNA gene sequencing (40 samples from 20 individuals).

Samples for shotgun metagenomic sequencing were collected using either D-Squame discs or OMNIgene SKIN kits. DNA extraction was then performed using one of the following methods: (i) the DNeasy PowerSoil kit, (ii) the QIAamp PowerFecal kit, or (iii) an in-house protocol referred to as the MGP protocol. Samples for 16S rRNA gene sequencing were collected using swabs, and DNA was extracted with the PowerSoil kit (Fig. 1). A complete description of all collected samples and their processing is available in Table S1.

Precise quantification of total DNA using a protocol specifically optimized for low concentrations was performed to select the most effective sampling and extraction methods prior to metagenomic sequencing. Overall, DNA amounts were low (Q1 = 1.08 ng; median = 1.62 ng; Q3 = 2.97 ng, Table S1; Fig. S1). Statistical analysis was conducted to assess the impact of sampling and extraction protocols on DNA yield (Fig. 2). Although forehead samples yielded more DNA (3.34 ± 3.93 ng) than armpit samples (2.53 ± 3.13 ng), the difference was not statistically significant ($P = 0.12$, linear mixed model).

In contrast, the collection method significantly influenced DNA yield ($P = 0.018$), with higher yields from D-Squame (3.63 ± 4.27 ng) compared with OMNIgene SKIN (2.24 ± 2.53 ng). The DNA extraction protocol had the greatest effect ($P = 1.0 \times 10^{-7}$): the MGP protocol yielded substantially more DNA (5.20 ± 5.16 ng) than either the PowerSoil (1.61 ± 1.45 ng) or PowerFecal (1.99 ± 1.42 ng) kits.

These results indicate that the optimal combination for maximizing DNA yield involves sampling with D-Squame and extracting with the MGP protocol. Due to the poor performance of the PowerFecal and PowerSoil kits, shotgun metagenomic sequencing was performed only on 40 samples from 20 donors that were extracted using the MGP protocol and collected with either OMNIgene SKIN or D-Squame, despite the slightly higher yields obtained with the latter.

### Multiple displacement amplification before library preparation distorts metagenomic profiles

Although DNA yields were the highest with the MGP protocol, total quantities were only a few nanograms per sample (Q1 = 1.62 ng; median = 2.88 ng; Q3 = 6.30 ng). These are about an order of magnitude lower than the manufacturer's recommendations for preparing sequencing libraries. In such a configuration, the risk that library preparation fails or that the sequencing generates too few reads must be considered. Therefore, MDA was carried out, with the aim of increasing the quantity of DNA available and thus the chances of successful sequencing.

For each of the 40 samples extracted with the MGP protocol, libraries were generated with and without 15 min MDA (40 × 2 = 80 libraries, see Materials and Methods). Shotgun

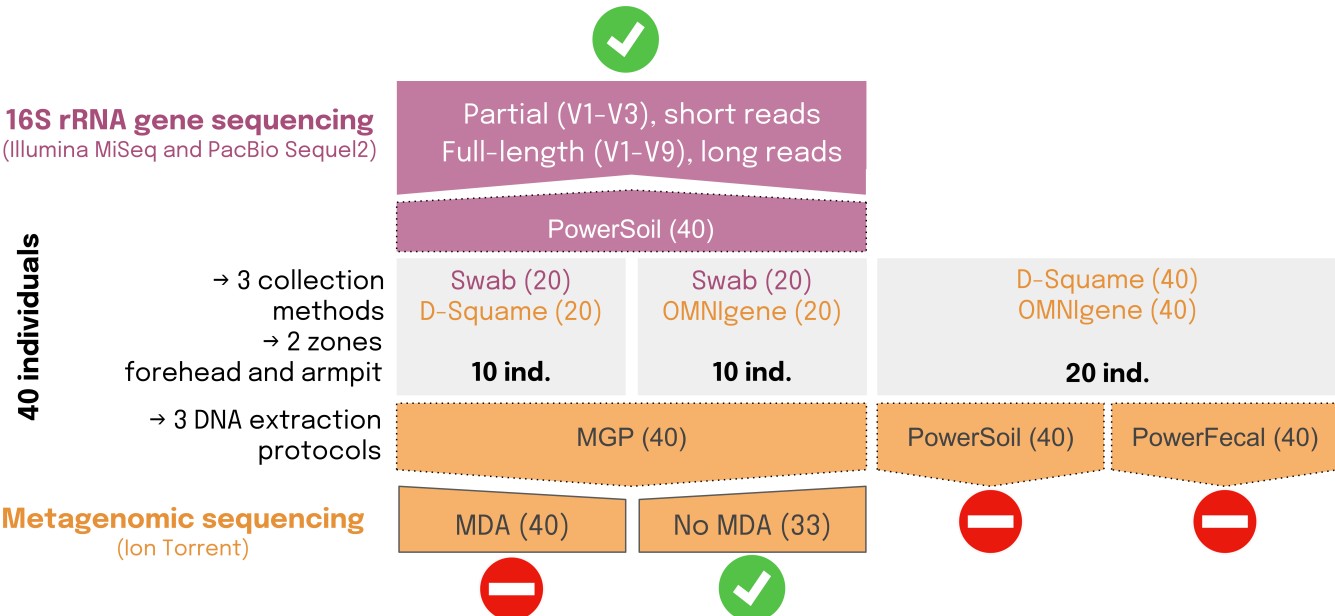

**FIG 1** Overview of the study design and experimental conditions tested for skin microbiome characterization. Forty healthy volunteers were divided into three groups, and skin samples were collected using three different methods: swab, D-Squame, and OMNIgene SKIN. Samples were taken from two body sites: the forehead and the armpit. Several DNA extraction protocols were tested: PowerSoil, PowerFecal, and an in-house protocol referred to as MGP. The study employed partial (V1–V3) and full-length (V1–V9) 16S rRNA gene sequencing (top section, purple text and boxes) as well as shotgun metagenomic sequencing (bottom section, orange text and boxes). For shotgun metagenomics, the impact of multiple displacement amplification was assessed, with some samples undergoing MDA and others not. Green check marks indicate the conditions included in the final analysis, whereas red symbols denote conditions excluded due to poor performance or quality issues.

sequencing after MDA was successful for all the samples, at least with regard to the volume of data generated: after quality control, at least 1 M reads were usable for all 40 samples (Q1 = 4.4 M reads; median = 20 M reads; Q3 = 22.5 M reads). As expected, sequencing without MDA was more problematic, and the yield was much lower (Q1 = 1.8 M reads; median = 3.2 M reads; Q3 = 9.5 M reads).

Among samples processed without MDA, a large proportion of reads mapped to the human genome, ranging from 3.6% to 98.2% (mean: 57.1% ± 32.4%). We observed no significant differences in host read proportions across body areas (armpit: 50.3% ± 36.5%; forehead: 63.8% ± 26.9%; $P$ = 0.23) or collection methods (D-Squame: 54.0% ± 36.6%; OMNIgene SKIN: 60.2 % ± 28.1%; $P$ = 0.56). Due to the high proportion of host-derived reads or low sequencing output, only 33 out of 40 samples (82.5%) without MDA reached the threshold of 1 million high-quality microbial reads (Table S2).

These preliminary results favored the use of MDA prior to library preparation, as it yielded better sequencing performance and allowed the inclusion of all samples in downstream biostatistical analyses. To rule out potential biases introduced by MDA, we focused on the 33 samples that reached at least 1 million microbial reads with and without MDA. For each sample, we compared the proportion of reads aligned to the human genome under both conditions. MDA significantly reduced the proportion of human reads (Q1 = +2.8%; median = −52.7%; Q3 = −90.4%; Wilcoxon signed-rank test, $P$ = 1.28 × $10^{-5}$), except in the most heavily contaminated samples.

Next, we generated taxonomic profiles for the same 33 samples, each processed with and without MDA, at a standardized sequencing depth of 1 million reads. Alpha diversity at the species level was assessed using both microbial richness (i.e., the number of species detected) and the Shannon diversity index (Fig. 3). Strikingly, with MDA, a significant decrease was observed in both species richness (average relative loss = −34.0%, $P$ = 1.73 × $10^{-5}$) and Shannon diversity (average relative loss = −16.9%, $P$ = 2.25 × $10^{-4}$).

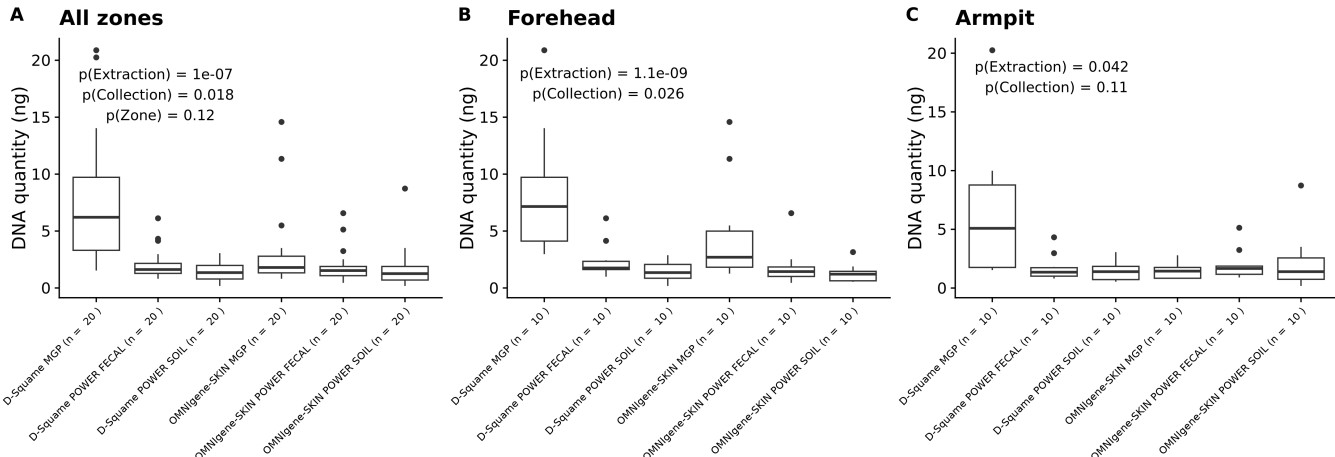

**FIG 2** Impact of different parameters (sample collection and extraction protocols) on DNA extraction yields considering all areas (A), only the forehead samples (B), or only the armpit samples (C). *P*-values from linear mixed models are shown.

In summary, MDA introduced major biases in microbial profiles, including a loss of diversity, which appeared too substantial and difficult to correct. Therefore, subsequent analyses were performed only on data generated without MDA (33 out of 40 samples). Among these non-MDA samples, more samples from the forehead met the quality threshold than those from the armpits (19/20 vs. 14/20, *P* = 0.09, Fisher's exact test), and more samples collected using D-Squame did so compared with those collected with OMNIgene·SKIN (18/20 vs. 15/20, *P* = 0.41). At least one sample was available for each of the 20 donors. For 13 donors, both a forehead and an armpit sample were available, whereas only one sampling site was available for the remaining seven donors (Table S2).

## Armpit and forehead harbor distinct skin microbial communities

From the species abundance table, we calculated a dissimilarity matrix between the 33 samples with the Bray-Curtis index. Permutational analysis of variance (PERMANOVA) revealed a major effect of the collected zone on the skin microbiome composition ($R^2$ = 0.34, *P*-value < 0.001), whereas the collection method played a negligible, non-significant role ($R^2$ = 0.02, *P*-value = 0.55, Fig. 4C). Based on this observation, we carried out analyses focused on the search for contrasts between the armpit and forehead microbiome.

Dissimilarity between all pairs of unrelated armpit samples was greater than that between unrelated pairs of forehead samples (mean Bray-Curtis dissimilarity: 0.75 *vs.* 0.45, *P*-value = $1.1 \times 10^{-9}$), suggesting that inter-individual diversity for the armpit microbiome is greater than for the forehead microbiome. No significant difference in microbial richness was observed between the armpit and the forehead (mean: 17 *vs.* 20 species, *P*-value = 0.74, Fig. 4B). However, the Shannon index, which takes into account the abundance of species, was higher in the armpit microbiome than in the forehead microbiome (mean 1.42 *vs.* 0.85, *P*-value = 0.0088, Fig. 4A).

Next, we explored sample composition at the genus and species level (Fig. 7). Overall, forehead samples were strongly dominated by the genus *Cutibacterium* (average abundance: 71% ± 34%). *Cutibacterium acnes* was the main contributing species (average abundance: 69.4% ± 33.4 %), but other less abundant species, such as *Cutibacterium granulosum* or *Cutibacterium namnetense,* were also detected (respective average abundance: 0.44% ± 0.93% and 0.96% ± 0.77%). Some donors exhibited an atypical forehead microbiome composition, such as that of donor 011, dominated by *Staphylococcus saccharolyticus* (abundance 38.1%) or that of donor 014, dominated by an uncharacterized species of the *Neisseriaceae* family. Yeasts of the *Malassezia* genus, represented by *Malassezia globosa* and *Malassezia restricta*, were frequent (3.7% ± 6.4 %). Of note, the DNA extraction protocol implemented in this study was optimized for

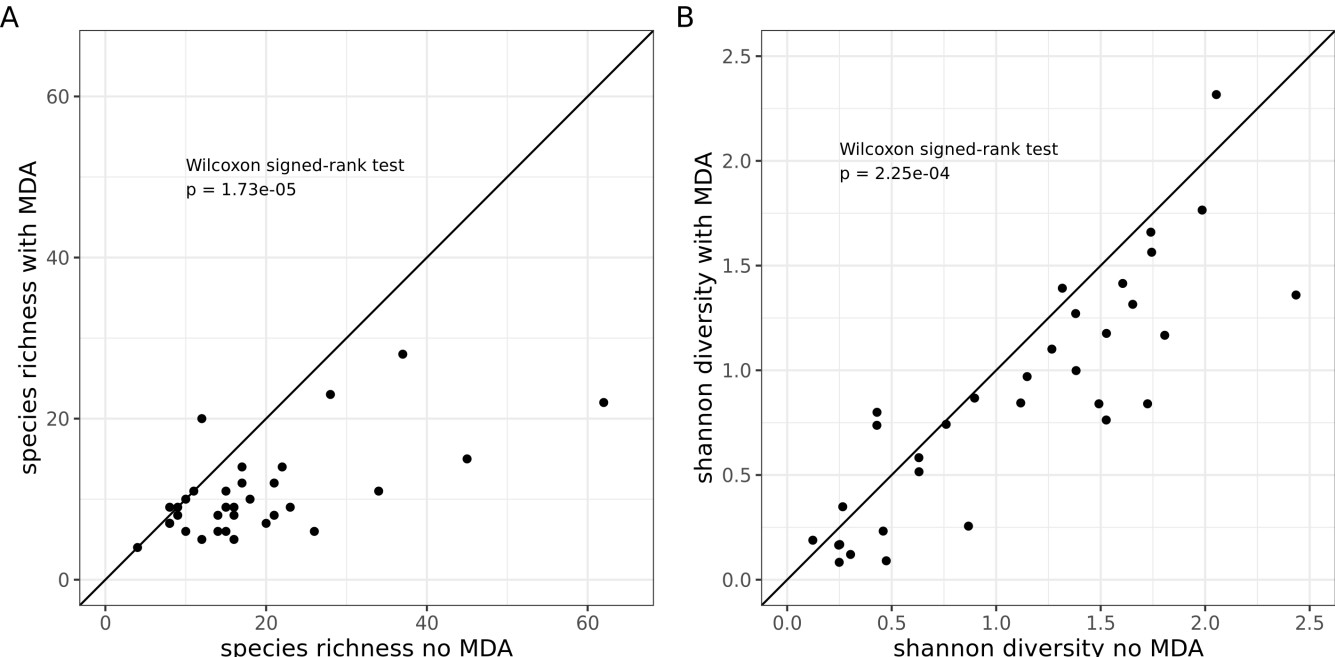

**FIG 3** Comparison of alpha-diversity in the 33 samples obtained with 15 min multiple displacement amplification (y-axis) or without amplification (x-axis) using (A) the microbial richness and (B) the Shannon index.

prokaryotes. Thus, although fungi were detected by sequencing, their abundance was likely underestimated (16).

The armpit samples were dominated by the *Staphylococcus* genus (average: 56.6% ± 25.7 %), *Staphylococcus epidermidis* (29.2% ± 23.8%), and *Staphylococcus hominis* (15.1% ± 22.8 %) being the main contributors. *Corynebacteria*, and more specifically the species *Corynebacterium aurimucosum*, *Corynebacterium kefirresidentii,* and *Corynebacterium haemomassiliense*, were also an abundant genus on the armpits (average abundance: 13.8% ± 19.3%), followed by *Anaerococcus spp.*. Notably, the dominance of the *Staphylococcus* genus was less pronounced than that of *Cutibacterium* for the forehead samples, which explains the difference in the Shannon diversity index reported by comparing these two zones.

Using nonparametric tests to identify differentially abundant species among those detected in at least five samples, we found 15 species with significant differences (false discovery rate [FDR] ≤ 0.1, Fig. 5A). This analysis confirmed that *C. acnes* and *S. epidermidi*s were typical species of the forehead microbiome (CD = −0.98, adjusted *P*-value = $5.1 \times 10^{-7}$) and the armpit microbiome (CD = 0.74, adj. *P*-value = $1.9 \times 10^{-3}$), respectively. We also confirmed the higher abundance of *Malassezia restricta* on the forehead (CD = −0.82, adj. *P*-value = $5.9 \times 10^{-4}$).

At the genus level, the results were consistent with those obtained at the species level. Among the 15 genera detected in at least 5 samples, eight were significantly enriched in one of the zones (FDR ≤ 0.1, Fig. 5B) including *Cutibacterium* (CD = −0.92, adj. *P*-value = $5.1 \times 10^{-6}$), *Staphylococcus* (CD = 0.80, adj. *P*-value = $6.0 \times 10^{-4}$), and *Malassezia* (CD = −0.82, adj. *P*-value = $3.8 \times 10^{-4}$). The uncultured genus *Neisseriaceae QFNR01* previously reported on the human skin (17) and in the nasal cavity (18) was also more abundant on the forehead (CD = −0.39, adj. *P*-value = $6.9 \times 10^{-2}$).

The skin virome was analyzed through the viral operational taxonomic units (vOTUs), which correspond to DNA sequences belonging to "species" of viruses (Fig. 5C). Of note, this analysis, based on data from whole metagenomes without prior virome enrichment, does not distinguish dormant phages (prophages) from active phages. Two hundred and thirty-nine vOTUs were detected in the 33 samples, mainly bacteriophages of the

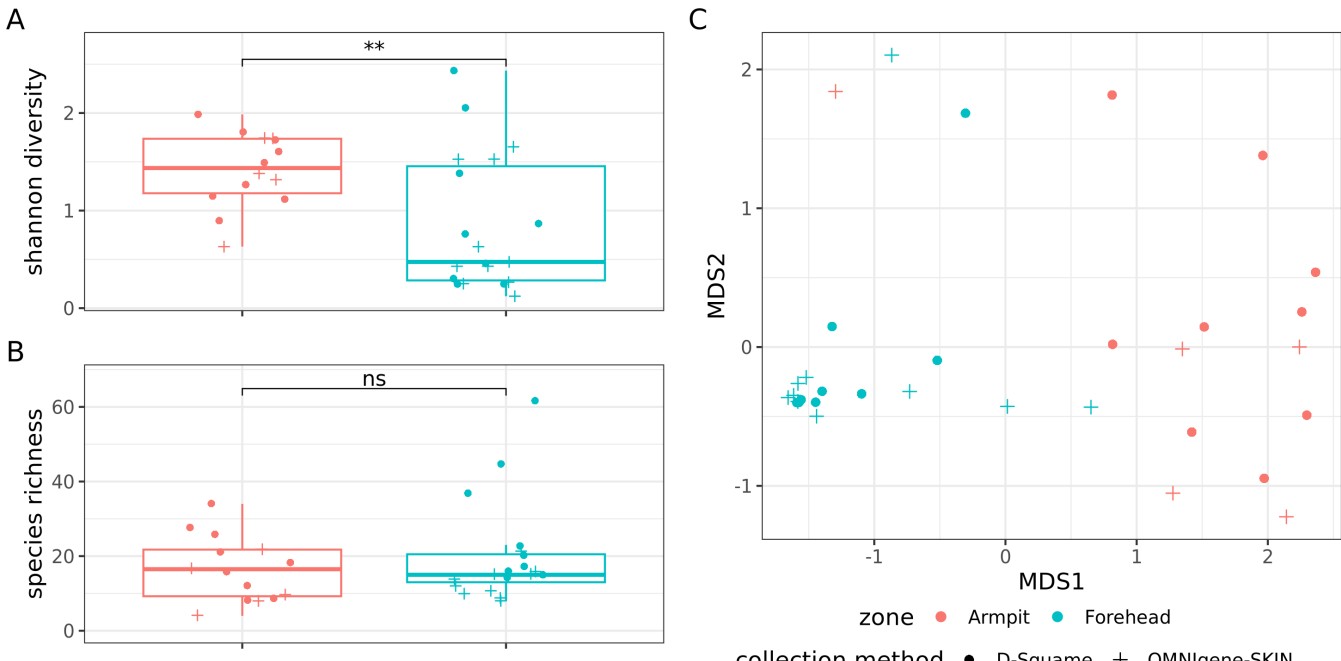

**FIG 4** (A) Shannon diversity index and (B) species richness are compared between zones. Statistical significance was assessed using Mann–Whitney U tests. Significance levels are indicated above the comparisons: **$P < .01$, ns = not significant. (C) Non-metric multidimensional scaling (NMDS) plot based on Bray–Curtis dissimilarities at the species level. Each point ($n = 33$) represents a sample. Colors indicate sampling zone (red: armpit, blue: forehead), and symbols indicate the collection method (dot: D-Squame; plus sign: OMNIgene·SKIN)

order Caudovirales (224/239), or tailed phages, as well as 11 papillomaviruses, specific to human cells and known to cause warts. Only 6 vOTUs were detected in at least 5 samples, suggesting a large interindividual variability in the composition of the skin virome. Eighty-three vOTUs were predicted to target *Staphylococcus spp*. and 19 *Cutibacterium spp*., which were the dominant genera on the armpits and forehead, respectively. Two phages targeting the *Cutibacterium* genus (vOTU_01430640 and vOTU_00221473) were the most abundant on the forehead, whereas two phages targeting the *Staphylococcus* genus (vOTU_02827251 and vOTU_08516476) were the most abundant on the armpits (Fig. 5C). This result suggests that the skin virome closely reflects the composition of the skin bacteriome.

## Skin microbial communities exhibit distinct functional potentials in the forehead and armpit

To provide a more comprehensive analysis of the microbiome, we investigated the functions of genes identified in the 33 samples from both areas. At the gene family level (KEGG Orthologs, KOs), the Shannon diversity index was similar between the forehead and armpits (mean = 7.45 vs. 7.43, $P$-value = 0.73, Fig. 6A). However, richness was significantly higher on the forehead (mean = 5,159 KOs vs. 3,837 KOs, $P$-value = 0.002, Fig. 6B), which was initially unexpected, given the lower species-level diversity in this area.

Consistent with the species-level analysis, a PERMANOVA test based on Bray-Curtis dissimilarity of KO abundances revealed a significant difference in microbial functional potential between the two areas ($R^2 = 0.33$, $P$-value < 0.001, Fig. 6C). Similarly, univariate analysis showed that nearly half of the KOs detected in at least five samples (3,701/7,248) exhibited significant differences between the two areas (FDR ≤ 0.1, Table S3).

Interestingly, KOs enriched in a given area frequently belonged to functionally related categories. For instance, 34 KOs related to porphyrin metabolism were enriched on the forehead, compared with only six in armpit samples. Given that porphyrins are well-established virulence factors of *C. acnes*, their enrichment in the forehead, where

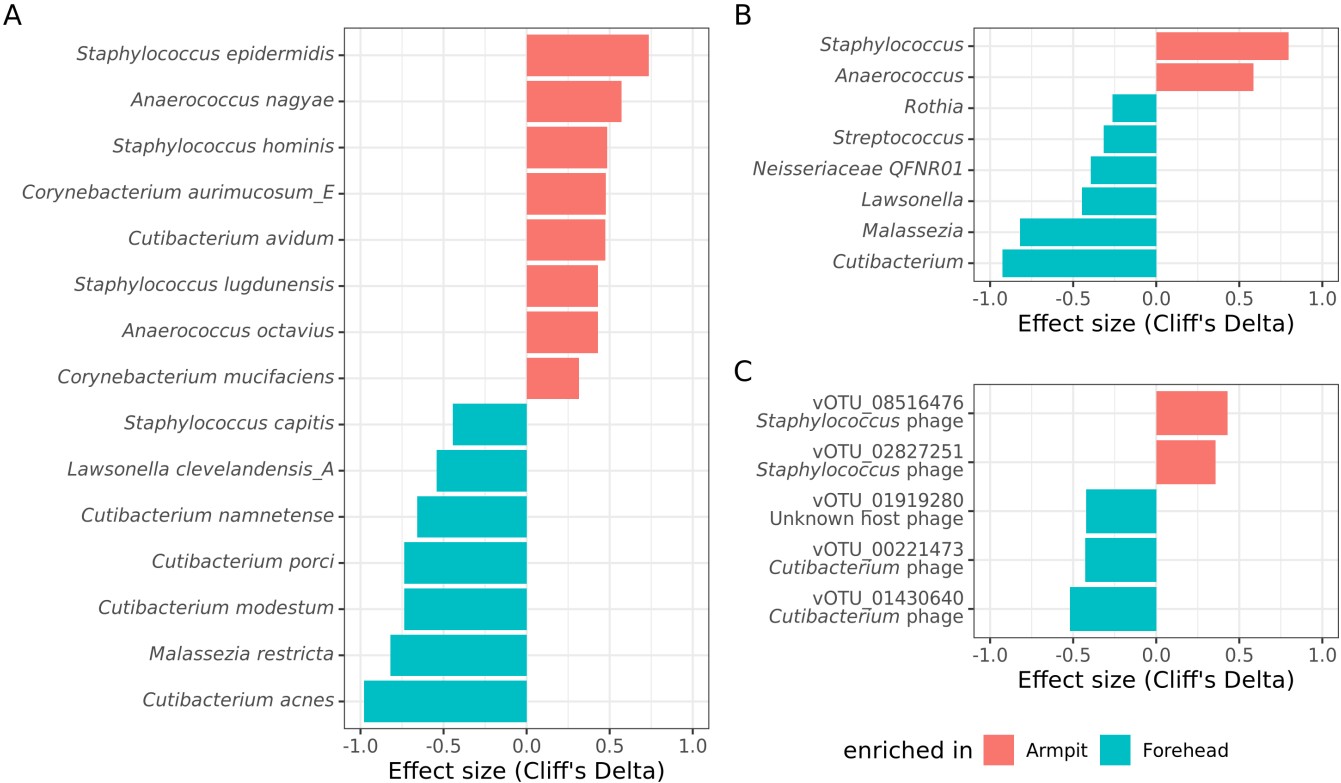

**FIG 5** Differentially abundant microbial species (A), genera (B), and vOTUs (C) between armpit and forehead samples. Horizontal bars represent the effect size (Cliff's Delta; range: −1.0 to +1.0), with positive values indicating enrichment in the armpit (red) and negative values indicating enrichment in the forehead (cyan).

this species is dominant, was expected. In addition to porphyrin-related functions, three other *C. acnes* virulence-associated KOs were significantly enriched in forehead samples: sialidase-1 (KO1186), hyaluronate lyase (KO1727), and the cAMP factor (KO11045).

Gene families associated with lipid metabolism were also enriched in forehead samples, with 117 lipid-related KOs identified compared with 52 in armpit samples, consistent with the sebaceous nature of the forehead. Notably, 15 KOs involved in ceramide and sphingolipid metabolism were specifically enriched in forehead samples, whereas none were detected in armpit samples.

Finally, three KOs associated with tryptophan metabolism were identified exclusively in forehead samples. Two of these were linked to the presence of *Malassezia spp*.

Specific KOs associated with the armpit region were less numerous. However, four KOs related to spermidine metabolism were found to be enriched in armpit samples, compared with just two in forehead samples. Notably, the KO1760 (cysteine-conjugate beta lyase, or patB) was also more abundant in armpit samples, with *S. epidermidis* and *S. hominis* being the primary contributors (45.6% and 21.4%, respectively).

Regarding antibiotic resistance genes identified by ResFinder, *blaZ* beta-lactamase and *fosB* fosfomycin resistance genes known to be carried by *Staphylococci* were more abundant in armpit samples (FDR ≤ 0.01).

## Comparison of shotgun metagenomics with partial and full-length 16S rRNA gene sequencing

As a reminder, for 20 individuals from the first two groups, two additional samples were collected from the forehead and armpits using swabs (Fig. 1). DNA was extracted using the DNEasy PowerSoil kit, and 16S rRNA gene sequencing was performed using two approaches: partial (V1–V3) with Illumina short reads or full length (V1–V9) with PacBio HiFi long reads. We considered the 33 samples successfully characterized by shotgun

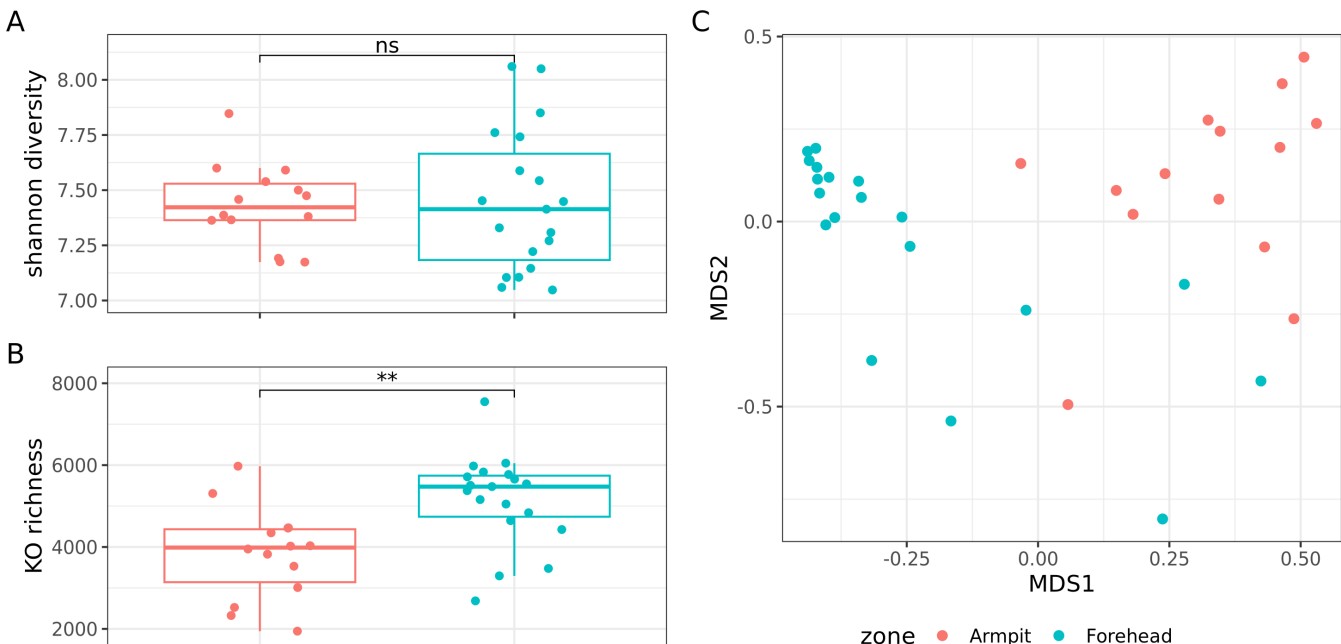

**FIG 6** Comparison of KO functional alpha-diversity between armpit and forehead samples using (A) the Shannon diversity index and (B) richness. Statistical significance is indicated above the comparisons: **$P < .01$, ns = not significant. (C) Non-metric multidimensional scaling (NMDS) projection of the Bray-Curtis dissimilarity matrix at the KO level. Each point ($n = 33$) represents a sample, with colors indicating body sites: red for armpits and blue for foreheads.

metagenomics and compared them with the corresponding 33 samples characterized by either partial or full-length 16S sequencing, collected from the same area on the same donor. The analysis was performed at the genus level (Fig. 7), the lowest taxonomic resolution achievable with partial 16S rRNA gene sequencing. Notably, a direct comparison of abundance profiles between the two sequencing technologies was not possible, as the two methods rely on different taxonomic reference databases (GTDB for shotgun metagenomics, SILVA for 16S rRNA gene sequencing). However, when relevant, we manually matched taxonomies between the data sets to enable the comparison shown in Fig. 7. Mantel tests based on Bray-Curtis dissimilarity matrices suggested that microbial compositions obtained with partial and full-length 16S rRNA gene sequencing were similar ($r = 0.98$; $P$-value $< 1 \times 10^{-4}$). In contrast, a notable dissimilarity was observed between samples processed by metagenomic sequencing and those processed by full-length ($r = 0.85$; $P < 1 \times 10^{-4}$) or partial 16S rRNA gene sequencing ($r = 0.8$; $P$-value $< 1 \times 10^{-4}$). Interestingly, this dissimilarity was more pronounced with armpit samples ($r = 0.53$ and $r = 0.65$) than with forehead samples ($r = 0.93$ and $r = 0.94$).

To better understand these site-specific differences, we compared the relative abundance of the most dominant genera, *Cutibacterium* and *Staphylococcus*, across the three sequencing techniques. Using the L1 norm, we found that *Cutibacterium* abundances were generally consistent across techniques (Fig. 8). However, in agreement with the Mantel test results, full-length and partial 16S sequencing showed greater similarity. Notably, *Cutibacterium* abundances from shotgun sequencing were more similar to those from full-length 16S than to those from partial 16S.

In contrast, *Staphylococcus* abundances varied substantially between 16S and shotgun sequencing (Fig. 8). Both full-length and partial 16S sequencing yielded higher *Staphylococcus* abundances compared with shotgun sequencing, with the discrepancy being the most pronounced on the forehead, where this genus is less abundant. These variations in *Staphylococcus* abundances likely explain the differences observed in the Mantel tests between forehead and armpit samples.

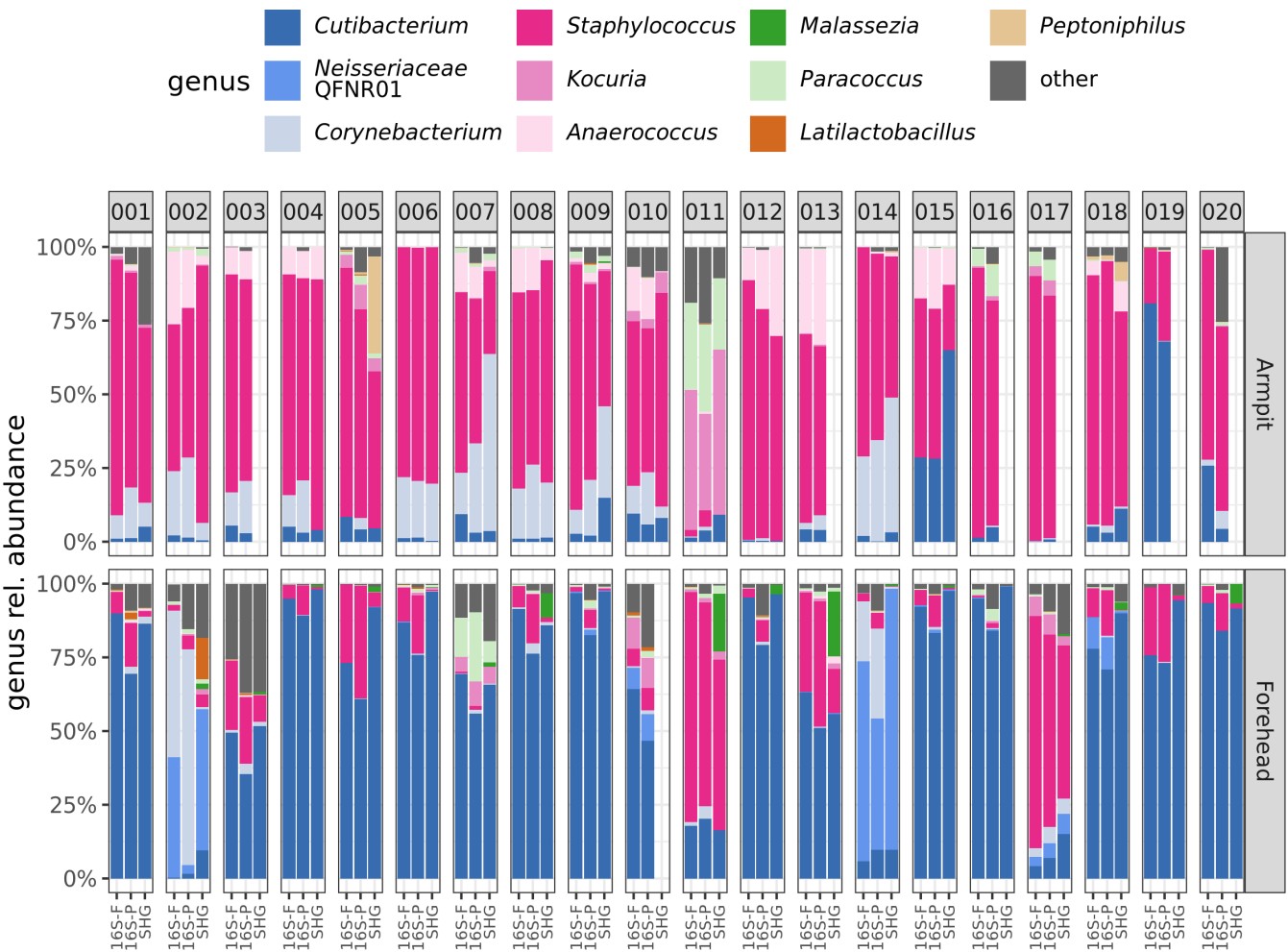

**FIG 7** Microbiome composition at the genus level across donors, body sites, and sequencing techniques. Stacked bar plots representing the relative abundance of microbial genera in skin samples collected from 20 donors (vertical panels) at two body sites: the armpit (top panel) and the forehead (bottom panel). For each donor and site, microbiome characterization was performed using three sequencing techniques: shotgun metagenomic sequencing (SHG), full-length 16S rRNA gene sequencing (16S-F), and partial 16S rRNA gene sequencing (16S-P). Colors represent different microbial genera, whereas the category "other" corresponds to the cumulative abundance of subdominant genera not assigned a specific color. Blank areas indicate missing shotgun metagenomics samples for the corresponding donor and site. Discrepancies in taxonomy between shotgun sequencing and 16S rRNA sequencing (e.g., *Neisseriaceae* unclassified vs. *Neisseriaceae* QFNR01, Lactobacillus vs. Latilactobacillus, were resolved by renaming some genera.

## DISCUSSION

This study aimed to optimize the methodology for analyzing the skin microbiome using shotgun metagenomic sequencing through the collection of samples from healthy volunteers. The results were then compared with those obtained from the canonical partial V1–V3 or full-length 16S rRNA gene sequencing approach.

DNA quantification after extraction indicated that D-Squame disks provided higher yields, suggesting that they were the most effective option for sample collection compared with OMNIgene·SKIN swabs. The adhesive surface of D-Squame disks likely maximizes microbial biomass collection without significantly increasing the proportion of human cells, at least on healthy, non-compromised skin.

Furthermore, the MGP in-house extraction protocol, which combines thermal, chemical, and mechanical lysis, resulted in higher DNA yields compared to commercial protocols. This may be due to its more aggressive approach to lysing resistant microbial cells. Regardless, these findings underscore the importance of testing and comparing

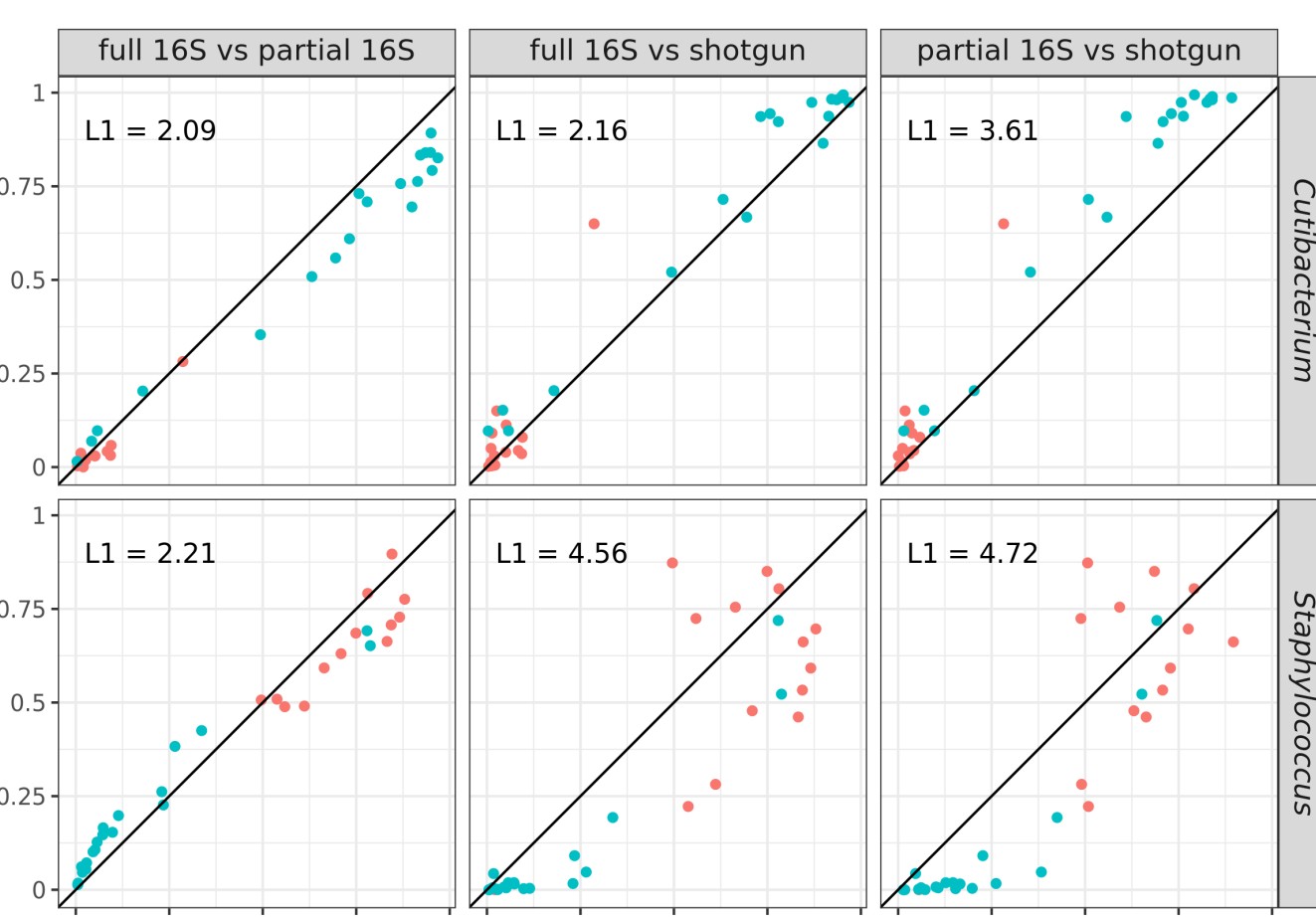

**FIG 8** Comparison of the relative abundance of the genera *Cutibacterium* (top row) and *Staphylococcus* (bottom row) across different sequencing approaches: full-length 16S vs. partial 16S (left column), full-length 16S vs. shotgun (middle column), and partial 16S vs. shotgun (right column). The L1 norm (Manhattan distance) is reported for each comparison. In each "X vs. Y" comparison, X is plotted on the x-axis and Y on the y-axis. Each point represents a sample, color-coded by anatomical site (orange = armpit, blue = forehead).

DNA extraction protocols before implementing them on a larger scale, especially for low-biomass samples.

Although the combination of carefully selected collection kits and optimized DNA extraction protocols improved yields, the total DNA quantities remained low, posing a risk of sequencing failure. To address this, we employed MDA to increase the likelihood of successful sequencing. As anticipated, the number of reads generated was significantly higher with MDA compared with non-amplified samples. However, MDA introduced substantial biases in taxonomic abundance profiles. Therefore, only non-amplified samples were analyzed, despite losing 7 samples due to insufficient microbial reads.

WGA techniques, such as MDA, are commonly used for shotgun metagenomic sequencing of low-biomass samples, but a few studies have assessed their impact. As previously reported, we confirm that WGA introduces significant biases and should be avoided unless there are no alternatives (19, 20). These biases may explain why some studies using WGA have reported fragmented metagenomic assemblies and a low number of high-quality metagenome-assembled genomes (MAGs), despite achieving high sequencing depths (21, 22).

Finally, we compared samples characterized by shotgun metagenomics with those subjected to 16S rRNA gene sequencing (both partial and full length). Notably, this comparison was not made between the same samples but rather between samples collected from the same donor and area, only a few centimeters apart. Therefore, it is reasonable to assume that the microbial composition of these "twin" samples is very similar. However, we observed a higher abundance of *Staphylococci* in the 16S rRNA gene sequencing results compared with shotgun sequencing.

These differences could be due to the sequencing methods themselves or variations in extraction protocols. *Staphylococcus spp*. have multiple copies of the 16S rRNA gene in their genome (9) (5 to 6 copies in *S. epidermidis* and *S. hominis*), which may lead, as previously reported (23), to an overestimation of the abundance of this genus relative to others with fewer copies, such as *Cutibacterium spp*. (three copies in *C. acnes*). Furthermore, different DNA extraction protocols were employed. The protocol optimized for 16S rRNA sequencing may have been more effective at lysing *Staphylococci*, which are gram-positive bacteria with thick, hard-to-lyse peptidoglycan cell walls (24). Although we intended to use the same extraction protocol for all sequencing strategies, the DNA yield from the PowerSoil kit was insufficient for metagenomic sequencing.

As previously reported, forehead and armpit microbiomes differed strongly, reflecting physiological differences between these areas. Despite a similar observed species richness in both zones, the Shannon index, which considers both richness and evenness, was significantly greater in the armpit samples. This underscores the strong dominance of *Cutibacterium acnes* on the forehead, whereas the armpit, although dominated by *Staphylococci*, exhibits a more even species distribution.

Interestingly, the opposite trend was observed when analyzing KEGG Ortholog abundance profiles. The Shannon index was similar between the two areas, but KO richness was significantly higher on the forehead. Further investigation revealed that this discrepancy was primarily driven by the presence of *Malassezia* spp. on the forehead, as these eukaryotes possess a functional repertoire distinct from that of bacteria. These findings also suggest some functional redundancy among the bacterial species of the skin microbiome.

*C. acnes* is nowadays considered a normal member of the skin microbiome, particularly abundant in sebaceous areas (25), as confirmed in this work. However, specific strains (or phylotypes) have been linked to the development of *acne vulgaris*. In this study, we identified several gene families in forehead samples associated with virulence factors of *C. acnes*, mainly porphyrins (36 KOs enriched on forehead), but also a CAMP factor, a sialidase, and a hyaluronidase. Porphyrins are inflammation activators in acne lesions and are produced in higher quantities by strains belonging to the IA1 phylotype (26, 27). Sialidase, hyaluronate lyase, and CAMP factors are damage-associated molecular pattern (DAMP)-inducing factors that degrade the extracellular matrix and favor the spread of bacteria in the tissues (28). It is noteworthy that these virulence factors were detected on healthy skin samples, and it would be interesting to compare their prevalence on acne skin and investigate their link with the *C. acnes* phylotypes.

Yeasts of the *Malassezia* genus, specifically *M. globosa* and *M. restricta*, were exclusively detected in forehead samples. *Malassezia* species require lipids for growth and most possess lipase-encoding genes (29), enabling them to thrive on sebaceous areas such as the forehead, shoulders, back, and scalp. Gene families involved in lipid metabolism were generally more abundant in forehead samples, including all those associated with ceramide metabolism, where *Malassezia restricta* was the main contributor. Ceramides are major components of the skin, contributing to its barrier function (30). Although previous studies have highlighted the role of *C. acnes* and *S. epidermidis* in ceramide synthesis, our study underscores the need to further investigate the role of *Malassezia* in ceramide metabolism.

In the presence of tryptophan, a component of sweat, *Malassezia* spp. produce indolic compounds (31) that can bind the aryl-hydrocarbon receptor (AhR). When activated, Ahr exerts pro-inflammatory effects (32), upregulates barrier-related proteins,

and accelerates keratinocyte differentiation (33). In this study, differentially abundant KOs related to tryptophan metabolism were consistently enriched in forehead samples. As expected, not only were Malassezia species involved but also *C. acnes* and another uncharacterized genus. It would be interesting to investigate whether such KOs are overrepresented in subjects with seborrheic dermatitis, a dermatological condition linked with the proliferation of *Malassezia* species.

Regarding the functional potential of the armpit microbiome, we found that the cysteine-conjugate beta lyase, or patB (KO1760), was enriched in samples from this area. This enzyme plays a well-known role in armpit malodors (osmidrosis) by converting S-Cys-Gly-3M3SH into the thioalcohol methyl-3-sulfanylhexan-1-ol (3M3SH), a key contributor to body odor. PatB was primarily identified in strains of *Staphylococcus hominis* from the axillary microbiome (34), which is consistent with the species linked with *patB* in this study, namely *S. epidermidis* and *S. hominis*.

In conclusion, this work confirms the potential of shotgun metagenomic sequencing for an in-depth analysis of the skin microbiome. It underscores the importance of optimized protocols for low-biomass samples, improving the reliability of shotgun sequencing and paving the way for more robust clinical studies focused on the skin microbiome.

## MATERIALS AND METHODS

### Volunteer recruitment

This single-center, open, controlled study included both intra- and inter-subject comparisons. It was conducted at the Intertek investigation center (Center d'Études Cliniques, Paris, France). Forty healthy volunteers (male and female), aged 21–49 years (mean age: 36 years), with Fitzpatrick skin phototypes II–V, were enrolled (Table 1). Written informed consent was obtained from all participants (c.f. Ethics declaration section). Individuals with tattoos, scars, moles, sunburn, or wounds on the forehead or armpit were excluded. Those with very oily or very dry skin were also not included. Participants were instructed not to shave within 48 h prior to sampling and to avoid applying any topical products from the evening before sampling.

### Sample collection

Three sampling methodologies were employed, with all procedures conducted under sterile airflow conditions generated by a portable hood.

TABLE 1   Panel description

|  | Number of subjects | Percentage |
|---|---|---|
| Sex |  |  |
| Female | 36 | 90% |
| Male | 4 | 10% |
| Phototype |  |  |
| I | 0 | 0% |
| II | 8 | 20% |
| III | 26 | 65% |
| IV | 4 | 10% |
| V | 2 | 5% |
| Face skin type |  |  |
| Combination | 23 | 58% |
| Oily | 1 | 3% |
| Dry | 2 | 5% |
| Normal | 14 | 35% |

For 16S rRNA gene sequencing, 40 samples (20 from the forehead and 20 from the armpits) were collected using sterile cotton-tipped swabs moistened with a sterile solution of deionized water containing 0.15 M NaCl and 0.1% Tween 20, a method widely used for skin microbiome sampling. Each swab was rubbed for 30 s over a 4 cm² area. After sampling, the swab tips were placed in sterile microtubes and stored at −80°C until further analysis.

Sixty samples were collected using OMNIgene SKIN kits (DNA Genotek, Ottawa, Canada) (30 from the forehead and 30 from the armpits), following the manufacturer's instructions. Sampling was performed on a 4 cm² area, after which the collection device was transferred to its accompanying tube containing the stabilization solution. Samples were stored at −20°C until further laboratory processing.

Sixty D-Squames, or corneodiscs (D-SQUAME D100; diameter 2,2 cm; Monaderm, Monaco) were collected by stripping until saturation of the disk (30 on the armpits, 30 on the forehead). Tape strips were placed in microtubes, with the sampling surface toward the inside of the tube, and stored at - 80°C until further analysis.

The sampling design, illustrated in Fig. 1, was as follows:

- Group 1 (10 subjects):
    - 2 samples from the forehead: one D-Squame and one swab
    - 2 samples from one armpit: one D-Squame and one swab
- Group 2 (10 subjects):
    - 2 samples from the forehead: one OMNIgene·SKIN kit and one swab
    - 2 samples from one armpit: one OMNIgene·SKIN kit and one swab
- Group 3 (20 subjects):
    - 2 samples from the forehead: one OMNIgene·SKIN kit and one D-Squame
    - 2 samples from one armpit: one OMNIgene·SKIN kit and one D-Squame

Group 3 included twice as many subjects as groups 1 and 2 because it was used to evaluate two DNA extraction protocols (PowerFecal and PowerSoil), whereas the first two groups were used to assess only a single extraction protocol.

## DNA extraction and quality control

For samples intended for metagenomic sequencing, DNA extraction was carried out by the SAMBO platform (MetaGenoPolis, INRAe Jouy-en-Josas, France). Three extraction protocols were tested. Two were based on commercial kits, namely DNEasy PowerSoil and QIAamp PowerFecal (Qiagen, Hilden, Germany). The PowerSoil kit was selected in part because it is used by the 16S rRNA gene sequencing provider, facilitating methodological consistency. The PowerFecal kit was chosen based on the recommendation of DNA Genotek, the manufacturer of the OMNIgene·SKIN sample collection kit. The third protocol, later called « MGP », was inspired by the one described in SOP 07 V2 of the IHMS project (35, 36) and is used in production at SAMBO for processing of fecal samples. Given the low biomass collected, only one extraction protocol was tested for each sample. Forty samples were extracted with the MGP protocol (20 donors * two zones), 40 with the DNAEasy PowerSoil protocol (10 donors * two zones * two collection methods), and another 40 with the QIAamp PowerFecal protocol (10 donors *two zones *two collection methods). DNA quantification was carried out using a FilterMax F3 plate fluorescence reader (Molecular Devices, San Jose, CA, USA) and the Quant-iT dsDNA High-Sensitivity Assay Kit Q33120 (Invitrogen, Carlsbad, CA, USA), with 2 µL of each sample analyzed.

For the 40 samples (20 donors * two zones collected with swabs) intended for 16S rRNA gene sequencing, DNA extractions were performed by Metys INRAE Transfert (GeT-PlaGe core facility, INRAe Toulouse, France) using the DNeasy PowerSoil Pro kit (Qiagen, Hilden, Germany), following the manufacturer's instructions. A heating step was added between the addition of lysing buffer and the vortexing of the PowerPro Bead Tube. The quantity and purity of total DNA were evaluated with a Qubit 2.0

fluorometer using a Qubit TM DNA HS assay kit (Thermo Fisher Scientific, Waltham, MA, USA). DNA integrity and concentration were assessed by capillary electrophoresis using an automated CE Fragment AnalyzerTM system (Agilent Technologies, Santa Clara, CA, USA) with the DNF-464-0500_HS Large Fragment 50 Kb Analysis Kit. Extraction quality was also assessed by measuring the absorbance at 260 and 280 nm wavelengths, using a NanoDrop 8000 Spectrophotometer (Thermo Fisher Scientific, Waltham, MA, USA).

## Shotgun metagenomic sequencing

Shotgun metagenomic sequencing was performed by the MetaQuant platform (MetaGenoPolis, INRAe Jouy-en-Josas, France). For the 40 samples extracted with the MGP protocol, two types of library constructions were carried out: construction from amplified DNA and standard construction recommended by the manufacturer for DNA amounts greater than 50 ng.

## Multiple displacement amplification

To limit the effects of varying buffer volume between samples, the required volume of sample was diluted in the extraction elution buffer and then concentrated by evaporation with Concentrator Plus (Eppendorf, Hamburg, Germany). The amplification reaction was carried out directly in the plate on the dry pellets.

MDA was performed with the EquiPhi29 DNA Polymerase kit (Invitrogen, Carlsbad, CA, USA) using ExoRes Random Primer (Thermo Fisher Scientific, Waltham, MA, USA), a mixture of oligonucleotides representing all possible hexamer sequences. To reduce bias, the amplification time was limited to 15 min on 400 pg. The amplification product was denatured by heating at 95°C for 3 min, then rehybridized by slow return to room temperature in order to reduce the proportion of single-stranded DNA.

## Library preparation and sequencing

Libraries were constructed with the IonPlus fragment Library Kit (Thermo Fisher Scientific, Waltham, MA, USA) and the IonXpress barcodes for multiplexing. First, the DNA of each sample (40 with MDA and 40 without amplification) was fragmented by sonication with the E220 robot (Covaris, Woburn, MA), targeting a distribution centered on 150 bp. The construction of the libraries from the purified, fragmented DNA was then continued without size selection in order to optimize the final yield. After quantification of the libraries, several sequencing runs were carried out on Ion Torrent devices (Ion Proton and IonGeneStudio S5) (Thermo Fisher Scientific, Waltham, MA, USA), with the aim of obtaining at least 1M high-quality microbial reads per sample.

## 16S rRNA gene sequencing

16S rRNA gene sequencing was performed by Metys INRAE Transfert (GeT-PlaGe core facility, INRAe Toulouse, France).

## V1–V3 sequencing

The V1–V3 region was amplified from purified genomic DNA with the primers J7–27F (CT TTCCCTACACGACGCTCTTCCGATC- TAGAGTTTGATCCTGGCTCAG) and W104–533R (GGAG TTCAGACGTGTGCTCTTCCGATCT- TTACCGCGGCTGCTGGCAC) (37) using 35 amplification cycles with an annealing temperature of 65°C. Because MiSeq sequencing (Illumina, San Diego, CA, USA) enables paired 300 bp reads, the ends of each read overlap and can be stitched together to generate high-quality reads of the entire region in a single run. Single multiplexing was performed using a custom 6 bp index, which was added to the reverse primer during a second PCR with 12 cycles using forward primer (AATGATACG GCGACCACCGAGATCTACACTCTTTCCCTACACGAC) and reverse primer (CAAGCAGAAGAC GGCATACGAGAT-index-GTGACTGGAGTTCAGACGTGT). The resulting PCR products were purified and loaded onto the Illumina MiSeq cartridge according to the manufacturer's

instructions. The quality of the run was checked internally using PhiX, and then, each pair-end sequence was assigned to its sample with the help of the previously integrated index. On average, 60,821 reads (sd: 6,158) were obtained per sample.

## V1–V9 full-length sequencing

The 27F:AGRGTTYGATYMTGGCTCAG and 1492R:RGYTACCTTGTTACGACTT universal primer set (37) was used to amplify the full-length 16S rRNA gene from the genomic DNA extracted from each sample using 20 amplification cycles with an annealing temperature of 57°C. SMRTbell libraries were prepared from the amplified DNA by blunt-ligation according to the manufacturer's instructions (Pacific Biosciences, Menlo Park, CA, USA). At each step, DNA was quantified using the Qubit dsDNA HS Assay Kit (Thermo Fisher Scientific, Waltham, MA, USA), and amplicon size was assessed using the DNF-474-0500_HS NGS Fragment Analysis kit (Agilent Technologies, Santa Clara, CA, USA). Purification steps were performed using AMPure PB beads (Pacific Biosciences). Then, purified SMRTbell libraries were sequenced using Binding kit 3.1 and sequencing kit 2.0, by diffusion loading onto 1 SMRTcell on Sequel2 (Pacific Biosciences) instrument at 70 pM with a 0.8 h pre-extension and a 15 h movie. Circular consensus sequence (CCS) reads were generated from the raw PacBio sequencing data using the standard software tools provided by the manufacturer (Pacific Biosciences) with minPasses = 3 and minPredictedAccuracy = 0.999 in the SMRT Link software (release_12.0.0.177059). On average, 54,433 reads (sd: 14,160) were obtained per sample.

## Bioinformatics for metagenomics

### Sequencing data quality control

Sequencing data quality control was carried out with AlienTrimmer (38). First, sequencing adapters were eliminated, then poor-quality reads were shortened, and reads that were too short (< 45 bases) were discarded. Finally, reads aligned to the human reference genome (T2T-CHM13v2.0, GCF_009914755.1) with bowtie2 (39) were removed.

To remove biases related to variable sequencing depth, one million high-quality reads were randomly selected in each sample with seqtk (40). Samples that did not reach this threshold were discarded. Reads were subsequently shortened to 100 bases to limit alignment issues at the edges of genes and to take into account an increased error rate beyond this position.

### Taxonomic profiling

A species abundance table was generated with MetaPhlan4 (41) based on the vOct22 database. The table was converted to the GTDB r207 taxonomy (42) using a custom script that processes eukaryotes separately. A genus abundance table was calculated by summing the abundance of species from the same taxon.

### Viral sequences profiling

Viral sequences profiling was performed with sylph (43) (version 6.1) and the IMG/VR database (44) (version 2022-12-19_7.1). Sylph parameters were optimized for small genomes for both the sketching (parameter: -c 100) and profiling (parameter: --min-num-kmers 20) steps.

### Functional profiling

The abundance of KEGG gene families (KOs) was obtained with Meteor (version: 2.0.14) (45). First, selected high-quality reads were mapped with bowtie2 (39) to a gene catalog representative of the human skin microbiome, comprising 2.9 million genes (46). Alignments with nucleotide identity < 95% were discarded, and gene counts were computed with a two-step procedure previously described that handles multi-mapped reads. Then, raw gene counts were normalized according to gene length. Finally, KOs'

abundances were computed by summing the abundance of genes assigned to the same KO. The gene annotation at the KO level was obtained with KofamScan (47) relying on KEGG 107.

## Bioinformatics for 16S rRNA gene amplicon sequencing

Bioinformatics processing was performed using the FROGS tool (48). Briefly, primer sequences were removed with cutadapt (49). Sequence clustering was performed with SWARM (50) using the fastidious method (d = 1). Taxonomic annotation was performed by aligning cluster representatives with BLASTn (51) against the SILVA database v138.1 (52) restricted to bacteria.

## Statistics

Statistical analysis was carried out with the R software suite (version 4.4.1) (53) using the tidyverse (version 2.0.0) and vegan packages (version 2.6-6.1). Unless stated otherwise, unpaired comparisons between two conditions were performed with the nonparametric Mann-Whitney U test. Paired comparisons were performed with the nonparametric Wilcoxon signed-rank test. Permutational analysis of variance (PERMANOVA) was carried out with the adonis function (999 permutations). For multiple testing, FDR was controlled using the Benjamini-Hochberg method with a threshold set at 10%. Non-parametric effect size was measured with the Cliff's Delta (CD) statistic implemented in the effsize package (version 0.8.1).

## ACKNOWLEDGMENTS

This work was funded by Seppic and the MetaGenoPolis grant ANR-11-DPBS-0001.

We thank Clémence Genthon, Maxime Manno, and Zoé Hohmann (INRAE Transfert Metys) for performing 16S rRNA gene sequencing.

## AUTHOR AFFILIATIONS

[1]Université Paris-Saclay, INRAE, MGP, Jouy-en-Josas, France
[2]Mercurialis Biotech, Rochecorbon, France
[3]Intertek, Centre d'Études Cliniques, Paris, France
[4]Seppic, Research and Innovation, Castres, France

## AUTHOR ORCIDs

Florian Plaza Oñate  http://orcid.org/0000-0001-6036-0989

## FUNDING

| Funder | Grant(s) | Author(s) |
|---|---|---|
| Agence Nationale de la Recherche | ANR-11-DPBS-0001 | Marine Gilles |

## AUTHOR CONTRIBUTIONS

Florian Plaza Oñate, Investigation, Methodology, Writing – original draft | Benoît Quinquis, Investigation, Methodology | Florence Thirion, Data curation, Investigation, Methodology | Marine Gilles, Methodology, Writing – review and editing | Christian Morabito, Conceptualization, Methodology, Supervision | Karine Valeille, Conceptualization, Methodology, Resources | Richard Martin, Methodology, Supervision | Bérengère Guidet, Project administration | Catherine Kern, Project administration | Sophie Pécastaings, Project administration, Supervision, Writing – original draft

## DATA AVAILABILITY

Metagenomic and amplicon sequencing data supporting the findings of this study have been submitted to the EMBL-EBI's European Nucleotide Archive (ENA) and are available under the BioProject accession number PRJEB80549.

## ETHICS APPROVAL

The research was conducted in accordance with the Declaration of Helsinki. The study protocol was approved by the French Committee for the Protection of Persons (CPP Ile-de-France XI) registered under reference 2021-A03032-39. Written consent was obtained from all participants.

## ADDITIONAL FILES

The following material is available online.

### Supplemental Material

**Figures S1 and S2 (Spectrum01732-25-s0001.docx).** Figure S1: DNA quantity distribution in the samples.Figure S2: Number of reads in the samples, without (left) or with MDA (right).
**Table S1 (Spectrum01732-25-s0002.xlsx).** Overview of all collected samples, including group and individual identifiers, sampling site (forehead or armpit), collection method, DNA extraction protocol, sequencing technique, and measured DNA quantity when available.
**Table S2 (Spectrum01732-25-s0003.xlsx).** Metagenomic sequencing statistics
**Table S3 (Spectrum01732-25-s0004.xlsx).** Gene families (KOs) enriched in the armpit/forehead samples

### Open Peer Review

**PEER REVIEW HISTORY (review-history.pdf).** An accounting of the reviewer comments and feedback.

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
