## [Reviewer comments · Microbiology Spectrum]

Microbiology Spectrum

Assessment of protocols for characterization of the human skin microbiome using shotgun metagenomics and comparative analysis with 16S metabarcoding

Florian Plaza Oñate, Quinquis Benoit, Florence Thirion, Marine Gilles, Christian Morabito, Karine Valeille, Richard Martin, Bérengère Guidet, Catherine Kern, and Sophie Pécastaings

Corresponding Author(s): Florian Plaza Oñate, Institut National de Recherche pour l'Agriculture l'Alimentation et l'Environnement

Review Timeline:

Submission Date:	June 5, 2025
Editorial Decision:	August 26, 2025
Revision Received:	October 16, 2025
Accepted:	October 22, 2025

Editor: Jan Claesen

Reviewer(s): Disclosure of reviewer identity is with reference to reviewer comments included in decision letter(s). The following individuals involved in review of your submission have agreed to reveal their identity: Xinzhao Tong (Reviewer #1); Bärbel Ulrike Foessel (Reviewer #3)

Transaction Report:

DOI: <https://doi.org/10.1128/spectrum.01732-25>

Re: Spectrum01732-25 (Assessment of protocols for characterization of the human skin microbiome using shotgun metagenomics and comparative analysis with 16S metabarcoding)

Dear Dr. Florian Plaza Oñate:

Thank you for the privilege of reviewing your work. Below you will find my comments, instructions from the Spectrum editorial office, and the reviewer comments.

Thanks for submitting your research to Spectrum. Your work has now been evaluated by two independent Reviewers who are enthusiastic about your project (as am I). The Reviewers highlighted some comments and suggestions to help improve the manuscript and I would be happy to consider a revised version that addresses these items in a point-by-point manner. In particular, could you comment on potential controls that might have been included to account for contamination, and if possible provide matching taxonomy across different methods used.

Revision Guidelines

Sincerely,
Jan Claesen
Editor
Microbiology Spectrum

Reviewer #1 (Comments for the Author):

This study investigates different DNA extraction kits and amplification methods to improve DNA yield from skin samples. The results confirm the advantages of the MGP extraction protocol for DNA recovery and highlight the diversity loss associated with MDA for microbial characterization. The authors also compare shotgun metagenomics with partial and full 16S rRNA gene sequencing for taxonomic and functional profiling of the skin microbiome, revealing the highest similarity between full-length 16S and shotgun metagenomics. The manuscript is well-written; however, several points could be addressed to further strengthen the work.

Major Comments:

Abstract

The abstract is quite long. Please shorten it to focus on the key findings and take-home messages.

Introduction

Line 133: Please provide a reference for the claim that human host DNA can make up 99% of the sequences. This value seems unusually high for typical skin metagenome sequencing unless there was significant contamination from human host cells.

Lines 138-139: A comparison between amplicon sequencing and shotgun metagenomics is only robust when using a mock community with a known microbial composition. When using test samples, it is not possible to determine which method is "correct" since there is no standard reference.

Results

Lines 142-150: This content appears to be a duplication of information found in the Methods section. Please shorten or remove it.

Lines 358-360: It would be helpful to include an ordination plot (e.g., PCoA or NMDS) to visually represent the differences in microbial composition across body sites and sequencing methods. Please also specify the statistical tests used for this comparison.

Methods

Lines 542-551: Please explain why the three experimental groups had different numbers of subjects.

Minor Comment:

Please ensure that all microbial names at or below the family level are italicized throughout the main text, figures, and tables.

Reviewer #3 (Comments for the Author):

In the manuscript submitted by Onate et al. the microbiome of the forehead and armpits of 40 healthy volunteers are characterized using shotgun metagenomics as compared to 16S metabarcoding. Two (three including swabbing for the 16S libraries) sampling methods and three extraction methods are compared in order to assess strategies to improve sequencing success.

Although it's a well written manuscript that deals with the important issue of how to reliably sequence low biomass samples as - in this case - the skin microbiome in a preferably unbiased manner using shotgun metagenomics, I anyhow have some major concerns referring mostly to study/analysis design and lack of transparency in reporting on material and methods and QC. In my opinion those issues need to be fixed, if possible, prior to publication.

For details see the line by line comments below:

l. 20: Although I understand that "non-bacterial" addresses the fact that viral and fungal members of the skin flora aren't caught, this isn't fully true as archaea are ...

l. 135/136: Why hasn't also a dry area been included to possibly find out where are the limits?

l. 142ff & Fig.1: To be honest, I don't really understand the initial idea and planned distribution of samples on the different sampling and extraction methods taking two samples per person and location, while testing 2 sampling methods and 3 extraction methods ... Ideally one would want to compare everything to everything, I think. (Although I understand that a compromise is necessary in the one or other way.) Moreover, I don't understand at all why for 16S metabarcoding a different sampling strategy (swabbing) has been chosen.

Moreover (might just be me?) the sampling/analysis scheme doesn't really come clear at this point, neither from the description nor from the figure. (At least I only understood it when reading the material and methods section.) E.g. it can't be seen from the figure how samples from the same persons or from forehead and armpit are distributed.

Further, in general I wonder why two extraction kits for normally rather high microbial biomass samples (soil and fecal samples) have been tested, rather than ones for low biomass and/or "host-contaminated" samples.

Finally I have to note, that it's not at all ideal to compare the different sequencing methods based on different extraction methods (although this might not have been intended).

I. 163: Might be just not clear to me: What is Q1 and Q3?

I. 191: "15-minute MDA": Is this the standard? Or has the appropriate time (with respect to quantity?) been tested beforehand? And just to make sure: Has the MDA been performed from the same samples that were also sequenced without MDA?

I. 192: "(402 = 80)" -> Can't be true, do you mean 40x2?

I. 199-202: Does the proportion of host reads correspond to the individuals sampled from?

213: O.k. here it says that the comparison MDA - no MDA was done on the same samples.

262: "...their abundance was likely underestimated." - Why do you think so? Is there evidence, that proportions are normally higher and/or fungi not well caught by the extraction methods used?

300-301: I'd say it rather "depends" on the bacteriome, as to my knowledge specific bacteria have their specific phages.

I.345: Why swabs?

I.352: "because of many discrepancies in taxonomy" -> Why? Please explain! Isn't it possible to use same reference database to annotate shotgun metagenomics and 16S metabarcoding data?

I.359: As the armpit samples show a higher diversity, if I remember right, this makes sense, I think?

I.398: Add "microbial" before "biomass". + Is there a mechanistic/method inherent explanation for this?

I.403: There is something missing after "microbial" in the end of the sentence.

I.421: By the way: Why couldn't both types of libraries prepared from the same DNA sample? Was there too little material to split the samples? And if so, wouldn't it have been better to pool the two samples and do both libraries from the same pooled sample?

I.428-431: Variation in copy numbers per se should be equally relevant for the metagenomics libraries, I think? - I'd rather assume that higher copy numbers, so more copies in the original DNA extract, foster the amplification bias in the metabarcoding approach ...

I.432: Is there any (citable?) evidence for that?

I.456: Remove "l" from "bacterial", as it's supposed to be "bacteria"?

I.508: Add a space between "sequencing" and "for".

I.520: Just for curiosity: Is there a scale or something that somehow defines the different skin types?

I.526-527: I wonder how it works to sample someone under sterile airflow conditions?

I.533 and whole section: For instruments, consumables etc. at first appearance company name, city and country have to be given in brackets.

I.572: I wonder, if QC of so low amounts of DNA on a NanoDrop instrument makes sense. - To my knowledge NanoDrop isn't a reliable tool for DNA concentrations < 10 ng or so.

I.591: Change "Libraries" to "Library"?

I.592: What was the starting amount of DNA for the libraries?

I.605-606 and also later: Add references for the primers used.

I.607: Please add full PCR protocol.

I.607-609: Doesn't need to be explained further, if you just mention that paired-end sequencing was performed. However, the used sequencing kit should be named.

I.610: I guess the indexing primes were not "home made", but "custom" or "in house" or something similar?

I.616: I think, PhiX-DNA needs to be spiked in for higher diversity, as just 16S fragments are too similar to each other ... Please also give percentage of PhiX spike in!

I.620: Add full cycling protocol.

Complete workflow: Which kind of controls were included at which steps to control for contaminations during sampling and or processing?

I.636: "Trimmed" instead of "shortened"?

I.634-642: Was somehow assessed how well the community was covered when using one million reads?

I.662-666: Were the 16S libraries also randomized to a certain sequencing depth? And if so, which one? How well was the community covered by this? Why was the clustering still done based on OTUs and what was the threshold?

I. 668,669: versions of R and packages used?

We sincerely thank the reviewers for the time dedicated to evaluating our manuscript and for their constructive comments. We hope that the revised version, together with our detailed point-by-point responses, clarifies the raised issues and satisfactorily addresses the reviewers concerns.

Below, the line numbers refer to the document with tracked changes.

Editor

Could you comment on potential controls that might have been included to account for contamination ?

Unfortunately, we did not include negative controls in our study. However, we took multiple steps to assess potential contamination in our metagenomic sequencing data:

1. Cross-sample contamination: We used a computational tool that identifies cross-sample contamination without requiring negative controls (10.1101/2025.01.15.633153). No evidence of cross-contamination was detected, which is consistent with the fact that DNA extraction for each sample was performed using individual tubes.
2. Reagent-derived contaminants: We screened for highly prevalent but low-abundance taxa, which are often indicative of contaminants introduced through lab reagents. No such taxa were identified in our dataset.
3. Potential sample-specific contamination: Two forehead samples (SEP_047 and SEP_063) contained trace levels (<0.01% relative abundance) of species typically found in the gut microbiome (e.g., *Phocaeicola dorei*, *Alistipes putredinis*, *Prevotella copri*). While these could represent minor contamination given that our lab predominantly processes fecal samples, it is also possible that these taxa were transiently present on the forehead due to dispersal from hands or other environmental contact. Importantly, even if these observations are due to contamination, their low abundance makes it unlikely that they significantly impact downstream analyses.

Overall, although negative controls were not included, our analyses suggest that contamination is minimal and does not meaningfully affect the conclusions of the study.

if possible provide matching taxonomy across different methods used ?

To ensure consistency across methods, we used the default GTDB r207 taxonomy for shotgun metagenomic data and the default SILVA 138.1 taxonomy for 16S rRNA gene sequencing data when generating Figure 7 (genus-level composition). Minor adjustments were made in the SILVA taxonomy to harmonize genus naming across databases:

- Neisseriaceae unclassified → Neisseriaceae QFNR01
- Lactobacillus → Latilactobacillus

Reviewer #1 (Comments for the Author):

This study investigates different DNA extraction kits and amplification methods to improve DNA yield from skin samples. The results confirm the advantages of the MGP extraction protocol for DNA recovery and highlight the diversity loss associated with MDA for microbial characterization. The authors also compare shotgun metagenomics with partial and full 16S rRNA gene sequencing for taxonomic and functional profiling of the skin

microbiome, revealing the highest similarity between full-length 16S and shotgun metagenomics. The manuscript is well-written; however, several points could be addressed to further strengthen the work.

Major Comments:

Abstract

The abstract is quite long. Please shorten it to focus on the key findings and take-home messages.

We had initially followed the editor's recommendation to stay within the 250-word limit (our first version was 249 words). Nevertheless, we agree that the abstract could be more concise. We have therefore shortened it by about 20% to focus more clearly on the key findings and take-home messages.

Introduction

Line 133: Please provide a reference for the claim that human host DNA can make up 99% of the sequences. This value seems unusually high for typical skin metagenome sequencing unless there was significant contamination from human host cells.

While such high proportions are not the norm across all skin sites, they can indeed occur, particularly in samples from dry or peeling skin. Previous studies have reported host DNA content exceeding 99% (e.g., 10.1038/nature13786, Supplementary Table 1), which is in line with our own observations, where some samples contained >97% host DNA. In the revised manuscript, we have moderated our wording and now provide this reference to support the statement :

« such as dry or damaged skin, host contamination exceeds 90% of the sequencing data generated (15). » (line 104)

Lines 138-139: A comparison between amplicon sequencing and shotgun metagenomics is only robust when using a mock community with a known microbial composition. When using test samples, it is not possible to determine which method is "correct" since there is no standard reference.

We agree that mock communities with defined composition provide the most robust benchmark to assess DNA sequencing techniques. In this study, however, our primary goal was to compare their outputs on real skin samples rather than to determine which method provides results closest to reality. While mock communities are valuable, they differ markedly from skin samples (e.g., higher biomass, absence of host DNA, and higher DNA quality), which can limit their relevance in this context.

We clarified this in the revised manuscript:

« In parallel, samples were analyzed by 16S sequencing (partial V1-V3 and full length) to compare the outputs of both approaches on biologically relevant samples characterized by low microbial biomass and high levels of host DNA » (lines 109-110)

Results

Lines 142-150: This content appears to be a duplication of information found in the Methods section. Please shorten or remove it.

In the journal guidelines, the Results section must precede the Methods, so we felt it was necessary to briefly repeat certain methodological details to ensure clarity and readability. While we acknowledge some overlap, the Methods section provides the full details, whereas this description in the Results is intended to help readers follow the flow of the manuscript.

Lines 358-360: It would be helpful to include an ordination plot (e.g., PCoA or NMDS) to visually represent the differences in microbial composition across body sites and sequencing methods. Please also specify the statistical tests used for this comparison.

The ordination plot showing differences in microbial composition across body sites and collection methods based on metagenomic sequencing is already provided in Figure 4C. As described in the main text, Mantel tests based on Bray-Curtis dissimilarity matrices were used to compare sequencing methods. It was not possible to provide a combined ordination plot for 16S rRNA gene sequencing and metagenomic data, as the taxonomic abundance tables rely on different reference databases (16S = SILVA 138.1, metagenomics = GTDB r207).

We have clarified this in the revised manuscript:

« Notably, a direct comparison of abundance profiles between the two sequencing technologies was not possible, as the two methods rely on different taxonomic reference databases (GTDB for shotgun metagenomics, SILVA for 16S rRNA gene sequencing). However, when relevant, we manually matched taxonomies between the datasets to enable the comparison shown in Figure 7. » (lines 328-332)

Methods

Lines 542-551: Please explain why the three experimental groups had different numbers of subjects.

The first two experimental groups included 10 individuals each to assess the MGP extraction protocol. The third group included 20 individuals because it was used to evaluate two extraction protocols (PowerFecal and PowerSoil). In other words, this group is twice as large to allow testing of twice as many extraction methods.

We have clarified this in the methods section of the revised manuscript :

« Group 3 included twice as many subjects as Groups 1 and 2 because it was used to evaluate two DNA extraction protocols (PowerFecal and PowerSoil), whereas the first two groups were used to assess only a single extraction protocol. » (Lines 509-511)

Minor Comment:

Please ensure that all microbial names at or below the family level are italicized throughout the main text, figures, and tables

All microbial names at or below the family level are now italicized throughout the text, figures, and tables.

Reviewer #3 (Comments for the Author):

In the manuscript submitted by Onate et al. the microbiome of the forehead and armpits of 40 healthy volunteers are characterized using shotgun metagenomics as compared to 16S metabarcoding. Two (three including swabbing for the 16S libraries) sampling methods and three extraction methods are compared in order to assess strategies to improve sequencing success.

Although it's a well written manuscript that deals with the important issue of how to reliably sequence low biomass samples as - in this case - the skin microbiome in a preferably unbiased manner using shotgun metagenomics, I anyhow have some major concerns referring mostly to study/analysis design and lack of transparency in reporting on material and methods and QC. In my opinion those issues need to be fixed, if possible, prior to publication.

For details see the line by line comments below:

l. 20: Although I understand that "non-bacterial" addresses the fact that viral and fungal members of the skin flora aren't caught, this isn't fully true as archaea are ...

Thank you for this comment. We have updated the manuscript and replaced "it excludes non-bacterial taxa" with "it excludes non-prokaryotic taxa" to more accurately reflect that archaea are taken into account.

l. 135/136: Why hasn't also a dry area been included to possibly find out where are the limits?

Dry skin areas were not included because they typically exhibit low and highly variable microbial biomass across individuals. In addition, these areas tend to have higher levels of host DNA contamination, further complicating analysis.

l. 142ff & Fig.1: To be honest, I don't really understand the initial idea and planned distribution of samples on the different sampling and extraction methods taking two samples per person and location, while testing 2 sampling methods and 3 extraction methods ... Ideally one would want to compare everything to everything, I think. (Although I understand that a compromise is necessary in the one or other way.)

Ideally, a full factorial design including all combinations of sampling and extraction methods (i.e., six samples per site and per individual) would indeed be desirable. However, this was not feasible because of the limited biomass and sampling surface available at each skin site, which prevents collecting such a large number of samples per individual.

Moreover, I don't understand at all why for 16S metabarcoding a different sampling strategy (swabbing) has been chosen.

Thank you for pointing this out. We understand that using a different sampling strategy for 16S rRNA gene metabarcoding may appear inconsistent. However, swabbing is the most

common method, routinely applied in clinical and observational studies (doi : 10.1080/09546634.2025.2470379, 10.1080/09546634.2025.2470379, 10.1038/s41598-025-13870-y) and we used it here as a benchmark. In parallel, we also wanted to test two newer sampling techniques that the manufacturers present as valuable alternatives to swabbing, in order to assess their potential for shotgun metagenomics. This design allowed us to both maintain continuity with established practice and evaluate innovative approaches.

Moreover (might just be me?) the sampling/analysis scheme doesn't really come clear at this point, neither from the description nor from the figure. (At least I only understood it when reading the material and methods section.) E.g. it can't be seen from the figure how samples from the same persons or from forehead and armpit are distributed.

We did our best to concisely describe the experimental design in the Results section and to represent it in Figure 1. We acknowledge that the design is complex and may be difficult to grasp at first glance. To clarify it, we now provide an updated Supplementary Table 1 detailing all groups (and the individuals belonging to them), all collected samples (with corresponding individual, body site, and collection method), and their subsequent processing in the laboratory (DNA extraction and sequencing protocol). The existence of this table is now explicitly mentioned in the Results section where the experimental design is described.

Further, in general I wonder why two extraction kits for normally rather high microbial biomass samples (soil and fecal samples) have been tested, rather than ones for low biomass and/or "host-contaminated" samples.

We agree that some commercial kits are specifically designed for low-biomass or host-contaminated samples. Our choice to test the PowerSoil and PowerFecal kits was based on several factors:

- (i) The PowerFecal kit is recommended by DNA Genotek, the manufacturer of the OMNIgene·SKIN sample collection system we employed.
- (ii) the PowerSoil kit was used by our 16S rRNA gene sequencing provider, which would have facilitated direct comparison with shotgun metagenomic data (although this comparison was ultimately not possible in practice).
- (ii) both have been used in published studies analyzing the skin microbiome
- (iv) our technical staff were already experienced with these kits

We have clarified this in the methods section of the revised manuscript. Lines 516-519

Finally I have to note, that it's not at all ideal to compare the different sequencing methods based on different extraction methods (although this might not have been intended).

As noted in the discussion, our initial aim was to use the same extraction protocol across all sequencing strategies. However, the DNA yield obtained with the PowerSoil kit was insufficient for metagenomic sequencing, which required higher input. For this reason, we applied different extraction methods, acknowledging this as a limitation of the study.

« Although we intended to use the same extraction protocol for all sequencing strategies, the DNA yield from the PowerSoil kit was insufficient for metagenomic sequencing » lines 412-414

1. 163: Might be just not clear to me: What is Q1 and Q3?

Q1 and Q3 refer to the first and third quartiles, respectively. These are standard statistical notations used together with the median to describe data distribution.

1. 191: "15-minute MDA": Is this the standard? Or has the appropriate time (with respect to quantity?) been tested beforehand? And just to make sure: Has the MDA been performed from the same samples that were also sequenced without MDA?

The 15-minute MDA duration was selected as it was sufficient to reach the minimal DNA quantities recommended by the manufacturer for library construction. MDA was performed on the same starting material as now stated in the manuscript :

« For each the 40 samples extracted with the MGP protocol, libraries were generated with and without 15-minute MDA (40x2 = 80 libraries, see **Error! Reference source not found.** »).

1. 192: "(402 = 80)" -> Can't be true, do you mean 40x2?

Yes, thank you for pointing this out. This was indeed a typographical error and should read 40 × 2. The manuscript has been corrected accordingly in the revised version.

1. 199-202: Does the proportion of host reads correspond to the individuals sampled from?

Yes, the proportion of host reads was calculated per individual sample, so each value corresponds to the specific donor and site sampled.

213: O.k. here it says that the comparison MDA - no MDA was done on the same samples.

The comparison was performed on the same sample material, each processed with and without MDA. We have edited the manuscript to clarify this :

"Next, we generated taxonomic profiles for the same 33 samples, each processed with and without MDA, at a standardized sequencing depth of 1 million reads."

262: "...their abundance was likely underestimated." - Why do you think so? Is there evidence, that proportions are normally higher and/or fungi not well caught by the extraction methods used?

Indeed, several studies have shown that DNA extraction methods optimized for prokaryotes are generally less efficient for fungi, due to their more robust cell walls. We have added a relevant reference to support this point in the revised manuscript. Line 238

300-301: I'd say it rather "depends" on the bacteriome, as to my knowledge specific bacteria have their specific phages

Thank you for this suggestion. We chose not to use the term “depends”, as it implies a strict causal relationship that our data cannot fully support. Instead, we revised the sentence to read “closely reflects the composition of the skin bacteriome”, which we believe more accurately describes the strong association observed without overinterpreting causality.

1.352: "because of many discrepancies in taxonomy" -> Why? Please explain!

We acknowledge that the statement “because of many discrepancies in taxonomy” required clarification, as also noted by Reviewer 2. What we meant is that a direct comparison of 16S rRNA gene sequencing and metagenomic data was not possible, because the taxonomic abundance tables rely on different reference databases (16S = SILVA 138.1, metagenomics = GTDB r207). We have clarified this in the revised manuscript:

« Notably, a direct comparison of abundance profiles between the two sequencing technologies was not possible, as the two methods rely on different taxonomic reference databases (GTDB for shotgun metagenomics, SILVA for 16S rRNA gene sequencing). However, when relevant, we manually matched taxonomies between the datasets to enable the comparison shown in Figure 7. » lines 328-332

Isn't it possible to use same reference database to annotate shotgun metagenomics and 16S metabarcoding data?

Indeed, it is now possible to use a unified reference database. The recently developed Greengenes2 resource integrates genomic and 16S rRNA databases within a consistent taxonomy based on GTDB (<https://www.nature.com/articles/s41587-023-01845-1>). However, this resource was not available at the time we conducted our analyses.

1.359: As the armpit samples show a higher diversity, if I remember right, this makes sense, I think?

When comparing samples processed by 16S rRNA gene sequencing and metagenomic sequencing, we observed higher dissimilarity for armpit samples than for forehead samples. This difference could be partly related to the higher microbial diversity typically observed in armpit samples. However, as discussed later in the manuscript, it is mainly driven by discrepancies in *Staphylococcus* abundances between the two sequencing approaches.

1.398: Add "microbial" before "biomass". + Is there a mechanistic/method inherent explanation for this?

We have added the term “microbial” before “biomass” as suggested.

As explained in the manuscript, the higher microbial biomass obtained with D-Squame can be explained by its adhesive surface and by the sampling procedure itself, in which the disk is repeatedly pressed onto the skin until it becomes saturated, thereby collecting more material than a swab.

1.403: There is something missing after "microbial" in the end of the sentence.

Yes, thank you for pointing this out. The word “cells” was missing after “microbial”. This has been corrected in the revised manuscript.

1.421: By the way: Why couldn't both types of libraries prepared from the same DNA sample? Was there too little material to split the samples?

The DNA yield per sample was indeed low, and preparing two separate libraries (one for shotgun metagenomics and one for 16S rRNA gene sequencing) would have significantly increased the risk of failure. In addition, the two sequencing approaches were carried out in different laboratories (INRAE MetaGenoPolis and INRAE Transfert METYS) each specialized in one of the two technologies.

And if so, wouldn't it have been better to pool the two samples and do both libraries from the same pooled sample?

Pooling the two samples might have been possible if both sequencing methods had been performed in the same laboratory and from samples collected using identical protocols. However, this approach would have increased the risk of handling errors and cross-contamination.

1.428-431: Variation in copy numbers per se should be equally relevant for the metagenomics libraries, I think? - I'd rather assume that higher copy numbers, so more copies in the original DNA extract, foster the amplification bias in the metabarcoding approach ...

Not exactly. The 16S rRNA gene is known to exhibit variable copy numbers between species and even between strains of the same species (<https://doi.org/10.1371/journal.pone.0057923>). In contrast, the marker genes used by taxonomic profiling tools in metagenomics are typically single-copy genes, specifically selected to avoid such quantification biases.

1.432: Is there any (citable?) evidence for that?

Yes, Gram-positive bacteria, including *Staphylococci*, are known to be difficult to lyse due to their thick and resistant peptidoglycan cell wall.

We have clarified this point in the revised manuscript and added this citation. Lines 409-412

1.456: Remove "l" from "bacterial", as it's supposed to be "bacteria"?

We corrected this typo. Thank you for noticing.

1.508: Add a space between "sequencing" and "for".

We added the missing space in the revised manuscript.

1.520: Just for curiosity: Is there a scale or something that somehow defines the different skin types?

Yes. As mentioned in the Methods section (Volunteer recruitment), we refer to the Fitzpatrick skin phototype classification, which categorizes skin types from I to VI according to their color and sensitivity to ultraviolet radiation

1.526-527: I wonder how it works to sample someone under sterile airflow conditions?

The Intertek investigation center used a portable hood similar to this one
<https://www.terrauniversal.com/portable-cleanbooth.html>
We added this information in the revised manuscript. Line 483

1.533 and whole section: For instruments, consumables etc. at first appearance company name, city and country have to be given in brackets.

We have added company names, cities, and countries at first mention for all instruments and consumables, as requested.

1.572: I wonder, if QC of so low amounts of DNA on a NanoDrop instrument makes sense. - To my knowledge NanoDrop isn't a reliable tool for DNA concentrations < 10 ng or so.

Yes, we agree that NanoDrop is not reliable for low DNA concentrations. These measurements were part of the routine QC performed by the service provider for 16S rRNA gene sequencing. However, we did not use these values in our analyses.

For shotgun metagenomic sequencing, we employed a dedicated protocol optimized for low DNA concentrations. This information was missing in the original version of the manuscript, and we have now added it in the revised version (lines 525-528)

1.591: Change "Libraries" to "Library"?

We did the change as suggested.

1.592: What was the starting amount of DNA for the libraries?

We have added the starting DNA amounts used for library preparation in the updated Supplementary Table 1.

1.605-606 and also later: Add references for the primers used

We added a reference as suggested.

1.607: Please add full PCR protocol.

1.607-609: Doesn't need to be explained further, if you just mention that paired-end sequencing was performed. However, the used sequencing kit should be named.

1.620: Add full cycling protocol.

Unfortunately, the provider performing the 16S rRNA gene sequencing has since closed, so we were unable to obtain these informations.

1.610: I guess the indexing primes were not "home made", but "custom" or "in house" or something similar?

We did the change as suggested.

1.616: I think, PhiX-DNA needs to be spiked in for higher diversity, as just 16S fragments are too similar to each other ... Please also give percentage of PhiX spike in!

To the best of our knowledge, PhiX spike-in was not used for PacBio full-length 16S rRNA gene sequencing. For Illumina short-read 16S sequencing, we were unable to obtain the exact percentage of PhiX.

Complete workflow: Which kind of controls were included at which steps to control for contaminations during sampling and or processing?

We did not include negative controls in our study. However, we took multiple steps to assess potential contamination in our metagenomic sequencing data:

1. Cross-sample contamination: We used a computational tool that identifies cross-sample contamination without requiring negative controls (10.1101/2025.01.15.633153). No evidence of cross-contamination was detected, which is consistent with the fact that DNA extraction for each sample was performed using individual tubes.
2. Reagent-derived contaminants: We screened for highly prevalent but low-abundance taxa, which are often indicative of contaminants introduced through lab reagents. No such taxa were identified in our dataset.
3. Potential sample-specific contamination: Two forehead samples (SEP_047_a and SEP_063_a) contained trace levels (<0.01% relative abundance) of species typically found in the gut microbiome (e.g., *Phocaeicola dorei*, *Alistipes putredinis*, *Prevotella copri*). While these could represent minor contamination—given that our lab predominantly processes fecal samples—it is also possible that these taxa were transiently present on the forehead due to dispersal from hands or other environmental contact. Importantly, even if these observations are due to contamination, their extremely low abundance makes it unlikely that they significantly impact downstream analyses.

Overall, although negative controls were not included, our analyses suggest that contamination is minimal and does not meaningfully affect the conclusions of the study.

1.636: "Trimmed" instead of "shortened"?

We replace « shortened » by « trimmed » in the revised manuscript as suggested.

1.634-642: Was somehow assessed how well the community was covered when using one million reads?

We assessed coverage indirectly by examining the relative abundance of detected species (non-zero abundances). The first decile was 0.03%, suggesting that one million reads is sufficient to capture even low-abundance species,

1.662-666: Were the 16S libraries also randomized to a certain sequencing depth?

And if so, which one? How well was the community covered by this?

All 16S reads were processed to generate taxonomic abundance tables.

For partial-length 16S, an average of 60,821 reads (SD: 6,158) were obtained per sample, while for full-length 16S, the average was 54,433 reads (SD: 14,160) per sample. We added this information in the revised manuscript.

Several studies indicate that 10,000–15,000 reads are sufficient to characterize complex bacterial communities (10.1038/sdata.2019.7), suggesting that our sequencing depth was adequate for capturing the majority of taxa present.

Why was the clustering still done based on OTUs and what was the threshold?

Although FROGS historically referred to OTUs, the pipeline actually produces ASVs because it uses the Swarm algorithm with the fastidious method ($d = 1$), combined with chimera removal and cluster filtering. This approach effectively denoises sequences and generates high-resolution sequence variants, so no conventional OTU clustering threshold (e.g., 97%) was applied. The terminology in the pipeline was updated accordingly in recent version of FROGS (see: <https://frogs.toulouse.inrae.fr/news/FROGSNews-June2023.html#FROGS-produces-ASV>)

We clarified this in the revised manuscript (lines 635-638)

l. 668,669: versions of R and packages used?

We added this information in the revised manuscript (lines 540-541)

Re: Spectrum01732-25R1 (Assessment of protocols for characterization of the human skin microbiome using shotgun metagenomics and comparative analysis with 16S metabarcoding)

Dear Dr. Florian Plaza Oñate:

Thank you for carefully addressing the Reviewers' comments from your previous submission. I am pleased to inform you that your manuscript has been accepted for publication in Spectrum!

Your manuscript has been accepted, and I am forwarding it to the ASM production staff for publication. Your paper will first be checked to make sure all elements meet the technical requirements. ASM staff will contact you if anything needs to be revised before copyediting and production can begin. Otherwise, you will be notified when your proofs are ready to be viewed.

Sincerely,
Jan Claesen
Editor
Microbiology Spectrum